# Correlation between Oral Lichen Planus and Viral Infections Other Than HCV: A Systematic Review

**DOI:** 10.3390/jcm11185487

**Published:** 2022-09-19

**Authors:** Alberta Lucchese, Dario Di Stasio, Antonio Romano, Fausto Fiori, Guido Paolo De Felice, Carlo Lajolo, Rosario Serpico, Francesco Cecchetti, Massimo Petruzzi

**Affiliations:** 1Multidisciplinary Department of Medical-Surgical and Dental Specialties, University of Campania “Luigi Vanvitelli”, Via Luigi de Crecchio 6, 80138 Naples, Italy; 2Independent Researcher, 81100 Caserta, Italy; 3Head and Neck Department, School of Dentistry, Fondazione Policlinico Universitario A. Gemelli–IRCCS”, Università Cattolica del Sacro Cuore, Largo A. Gemelli, 8, 00168 Rome, Italy; 4Department of Social Dentistry and Gnatological Rehabilitation, National institute Health, Migration and Poverty (NIHMP), 00153 Rome, Italy; 5Interdisciplinary Department of Medicine, University of Bari “Aldo Moro”, 70121 Bari, Italy; 6Department of Restorative, Preventive and Pediatric Dentistry, University of Bern, Freiburgstrasse 7, 3012 Bern, Switzerland

**Keywords:** oral lichen planus, HPV, EBV, HSV-1, CMV, oral cancer

## Abstract

Objectives: This review aimed to evaluate the correlation between viral infections (HPV, EBV, HSV-1, CMV) other than HCV and oral lichen planus to assess if there is sufficient evidence to establish if these viruses can play a role in the etiopathogenesis of the disease. Materials and methods: We reviewed the literature using different search engines (PubMed, ISI Web of Science, and the Cochrane Library), employing MeSH terms such as “oral lichen planus” and “OLP” in conjunction with other terms. We utilized the Population, Intervention, Comparison, Outcomes, and Study design (PICOS) method to define our study eligibility criteria. Results: A total of 43 articles of the 1219 results initially screened were included in the study. We allocated the 43 selected items into four groups, according to each related virus: HPV, EBV, HSV-1, and CMV. Conclusions: Heterogeneous results neither confirm nor exclude a direct correlation between the investigated viral infections and oral lichen planus etiopathogenesis and its feasible malignant transformation. Many viral agents can cause oral lesions and act as cancerizing agents. Future studies could be desirable to produce comparable statistical analyses and enhance the quantity and quality of the outcomes to promote the translation of research into clinical practice.

## 1. Introduction

Oral lichen planus (OLP) is a chronic mucocutaneous inflammatory disease affecting the oral mucosa [1,2]. It tends toward middle-aged female patients [3]. Clinically, oral lichen planus (OLP) can present six patterns: papule, reticular, plaque, atrophic, erosive, and bullous [4], each exhibiting distinctive features and appearing isolated or as associated forms. The erosive lesions show greater clinical significance as they are usually symptomatic, varying from slight discomfort to episodes of severe pain [1]. OLP manifestations can continue for years, alternating between phases of quiescence and exacerbation [2].

The follow-up of OLP patients has shown evidence of the disease’s malignant potential [3,5,6,7,8]. Despite the numerous factors associated with malignant change, there is a scarcity of definite clinical, histologic, and molecular predictors of malignant development for OLP [9].

Upon histopathological examination, OLP lesions can present hyperkeratosis, hydropic degeneration of the basal cell layer of the epithelium, and dense well-defined lymphocyte T infiltrate in the superficial subjacent connective tissue [5].

The etiology of OLP is debatable, but some evidence indicates that the disease is an immunological process started by an antigen that varies the basal keratinocytes of the oral mucosa, exposing them to cell immune responses. It generates the activation of CD4+ T and CD8+ T lymphocytes and cytokine production, such as interleukin-2 (IL-2), interferon-gamma (IFN-g), and tumor necrosis factor (TNF), which determine the keratinocytes apoptosis [1,2,10,11]. The antigen that triggers this inflammatory response is still unknown. It might have an intrinsic or an extrinsic origin [1].

Several reports have pointed out a possible association between OLP and viral infections. Human papillomavirus (HPV) and four human herpesvirus family subtypes have been associated with oral manifestations of lichen planus, such as herpesvirus [1,4,12], Epstein–Barr [12,13], cytomegalovirus [14], and HPV [15].

However, some doubts remain about whether these agents are associated with OLP or whether the infection superimposes the already existing lesions [10].

The most broadly investigated virus in the OLP etiology is the hepatitis-C virus (HCV). Several meta-analyses have demonstrated strong evidence of an association between HCV and OLP [16,17,18]. This association could be justified by the capacity of the HCV virus to infect other cells besides hepatocytes, such as epidermal cells. The virus’ high mutation rate results in repeated activation of the immune cells, raising the risk of developing autoimmune diseases [19].

This systematic review investigates available evidence between the viruses mentioned above (except HCV, in which the OLP correlation has already been widely demonstrated [1,16,17,18,19]) and OLP. In detail, this review aimed to:Assess the prevalence of the viruses mentioned above (except for HCV) in OLP lesions;Evaluate whether any clinical features correlate with the viruses;Evaluate if there is any relationship between malignant transformation in OLP and viral infection.

## 2. Materials and Methods

The method used in this systematic review was adapted from the Preferred Reporting Item for Systematic Reviews and Meta-Analyses (PRISMA) guidelines [20]. An electronic search in MEDLINE, EMBASE, Web of Science, and the Cochrane Library databases was performed for original studies published in English. No publication year limit was applied so that the search could include the first available until 19 April 2022. Table 1 shows the search terms (MeSH terms) utilized.

The population of interest consisted of men and women with or without oral lichen planus; the intervention was the investigation of the presence/absence of the abovementioned viruses in patients with OLP; the comparison was the investigation of the presence/absence of the abovementioned viruses in patients without OLP; the outcome was to evaluate the presence/absence of the abovementioned viruses in patients with or without OLP; and the study designs included randomized control trials (RCTs), prospective comparison studies, retrospective cohort studies, case-control studies, and case series.

Quality assessment of non-randomized studies was evaluated on the Risk of Bias in Non-randomized Studies of Interventions (ROBINS I) assessment tool [21] to assess seven bias domains, and each one refers to the Risk of Bias (RoB) in five grades: low, moderate, severe, critical, and no information.

This review was submitted and registered on PROSPERO (registration number: CRD42022315611). Table 2 and Table 3 summarize the results retrieved.

## 3. Results

The initial electronic search turned out 1219 results. Titles and abstracts derived from the investigation were independently screened by three authors (AR, FF, GPDF) based on the listed criteria. Comparing the labels of the initial 1219 results, 304 were duplicates, and 861 were eliminated after having screened the titles based on the listed exclusion criteria. Then, 54 abstracts were analyzed; full-text articles were obtained for all titles agreed upon, and disagreements were resolved by discussion. A total of 11 studies were excluded (1 full-text article was not available; 3 were not related to OLP but systemic LP affecting the skin and mucous membranes, but it was not specified if any oral lesions were tested; 1 was a theoretical study; and 6 did not meet the requirements for inclusion on closer reading). Finally, 43 studies were included in the present review. The flowchart in Figure 1 explains the steps made during the selection process. We summarized and schematized the included studies in two different tables (Table 2 and Table 3).

### 3.1. Risk of Bias in Individual Studies

When assessed with ROBINS-I, 20 studies were graded as moderate-risk RoB, 22 as low RoB and only one as serious-risk. Considering that in this review 43 non-randomized studies were analyzed, all the studies provide sound evidence and none presented a critical RoB in any domain.

### 3.2. HPV

Thirty-three studies investigated HPV detection in OLP lesions as reported in Table 2. The prevalence of different HPV genotypes in OLP lesions, the correlation between clinical features associated and the presence of the virus, and the relationship between malignant transformation in OLP and HPV infection were assessed for each study. Results obtained in the different studies varied a lot, with a range of detection that went from 0 to 100%. They were difficult to compare because most of these research works involved a small number of samples and many different viral detection techniques.

Fifteen articles correlated OLP clinical features with HPV presence (Table 2).

Kato et al. [43] reviewed the most extensive group of OLP patients and found an HPV DNA prevalence of 41.5% (83/200). The most analyzed and retrieved genotypes were the “high risk” 16 and 18. HPV16 positive rates according to OLP clinical type were 28.3%, 18.6%, 25.9%, and 0.0% for erosive, reticular, plaque-type, and atrophic.

Recent studies such as Della Vella et al. in 2021 report a total prevalence of HPV infection of 17%, with a prevalence of 19% in the hyperkeratotic OLP and 11% in the erosive forms [53]. In 2021, Kaewmaneenuan et al. [52] published a research article that investigated the presence of HPV16 and 18 in OPMD and its correlation with demographic, risk habits, and clinical features. The samples collected belonged to 59 OLP patients. HPV16 and 18 were detected in OLP samples with a prevalence of 18.6% and only in patients with atrophic/ulcerative features. Statistical analysis revealed no significant correlation between HPV prevalence and risk factors. In 2020, Farhadi et al. [51] analyzed 32 OLP samples (12 reticular and 20 erosive) and 20 healthy samples and performed PCR looking for HPV DNA. Eight of the OLP (25%) and none of the healthy samples tested positive for HPV DNA. The HPV DNA-positive samples were further tested for the specific primers, revealing in 7 samples the 300 bp band attributable to HPV 33, and one sample exhibited the 873 bp band attributed to HPV 18. All of these samples belonged to erosive forms. Statistical analysis showed that HPV presence was significantly higher in the OLP-positive pieces. In 2019, Sameera et al. [50] collected 30 samples, of which 15 had histological diagnoses of OLP and 15 showed healthy buccal mucosa. PCR was performed to amplify the HPV L1 gene. A total of 86.6% (13/15) of OLP and none of the healthy mucosa samples were HPV 18 positive. Among HPV+ OLP samples, 8/13 were non-atrophic and 5/13 were atrophic types. No significant relation was observed based on the lesion’s site and the type of lesion.

Moreover, Viguier et al. [46] presented an exciting strategy: they examined the specificity of lesional cytotoxic T-lymphocyte (CTL) in 10 severe erosive OLP patients. The authors demonstrated that a significant proportion of clonal CD8+ blood T cells of OLP patients are HPV16 specific, infiltrating lesions and disappearing after clinical remission. Considering that three out of six patients have a current or past HPV infection, the authors also speculate that HPV infection may be an antigenic stimulus of CTL expansion that characterizes severe erosive OLP [46,63,64,65,66].

The results obtained by Kashima et al. [25] are peculiar; they examined 22 oral lichen planus specimens with two different techniques, finding different results. At immunoperoxidase examination, 4 out of 22 OLP specimens were positive for HPV capsid antigen (all women and two men showed koilocytes); none were found positive for HPV DNA by in situ hybridization.

In a 2018 study by Liu et al. [47], 40 OLP patients and 6 patients with malignant transformed (MT) OLP were enrolled. Their specimens underwent immunohistochemical staining and optometric density evaluation compared with 24 normal healthy mucosae. Expression of HPV16/18 infection was significantly higher in MT-OLP specimens (100%) than in OLP specimens (67.5%) and normal mucosa (62.5%).

Mattila et al. [41] registered a prevalence of 15.9% (13/82) of HPV infection in OLP patients: five of these lesions transformed into cancer, though, in the original biopsies utilized for the study, no dysplasia was present. None of the five lesions which moved to invasive carcinoma were found positive for the “high risk” HPV genotypes, while they were found positive for the assumed “low risk” HPV 6 and 11 genotypes.

### 3.3. EBV

Twelve studies assessed EBV prevalence in tissue specimens (Table 3). Results were very equivocal, as the positivity range varied from 0% (0/22, 0/25) to 74% (73/99) as reported in Table 3. 

Several authors evaluated the virus’ influence on the immune status of OLP patients. Pedersen [54] conducted a case-control study, investigating a possible association between oral lichen planus and the humoral immune response of the patients to EBV using an optometric density. In 22 OLP patients (and 20 healthy controls), serum samples were taken, and specific serum IgG antibodies (Ab) towards EBV early antigen (EA) and nuclear antigen-I (EBNA) and IgM Ab towards EBV EA were determined by ELISA. The IgG anti-EA Ab levels were significantly higher in OLP patients than in controls (12:5), and a quite negative correlation between the duration of symptoms from OLP and IgG anti-EA Ab values was found.

Along the same lines as the study mentioned above, Adtani et al. [58] performed a qualitative analysis of EBV viral capsid antigen (EBV VCA) IgG in the sera of OLP patients. They collected blood samples from 25 OLP patients (and 25 healthy controls), finding positivity in 4 cases (16%) for EBV VCA IgG, all females.

Four articles correlated OLP clinical features with EBV presence.

Raybaud et al. [61] enrolled 99 patients with OLP diagnosis and 22 patients diagnosed with oral lichenoid lesions as controls. The authors performed EBER-ISH that detected an EBV prevalence ranging around 74% with a significantly higher frequency (83%) in the erosive form than in the reticular/keratinized type mild form (58%). The surrounding healthy tissue belonging to OLP samples tested negative for EBV presence.

Vieira Rda et al. [59] investigated the EBV DNA presence in saliva samples, exfoliated cells, and plasma obtained from 24 patients with OLP and 17 healthy controls. In saliva samples, EBV was found in 75% of cases and 64.7% of controls. In the exfoliated cell samples, 70.8 and 82.4% of the patients and controls were positive for EBV. For plasma, 33.3% and 47.1% of the cases and controls samples were positive for EBV. The non-atrophic-erosive variables were the most prevalent in the patients (54.2%) and were also the most affected by the EBV virus. However, there were no significant differences between the clinical variations of OLP in each source material.

Sand et al. [13] found a significant difference between EBV prevalence in OLP patients (6/23) and control subjects (5/67). Of the six EBV-positive OLP patients, two had atrophic OLP, three had reticular OLP, and one OLP was unspecified.

Kis et al. [57] enrolled 68 healthy subjects and 116 patients with OLP, divided into two sub-groups: 59 erosive atrophic OLP (EA-OLP) and 57 non-erosive atrophic OLP (non-EA-OLP). EBV prevalences were comparable in the lesion (45.6%, 26/57 in non-EA-OLP; 47.5%, 28/59 in EA-OLP) and on the adjacent healthy mucosa (33.3%, 19/57 in non-EA-OLP; 30.5%, 18/59 in EA-OLP).

No article evaluated if there is any relationship between malignant transformation in OLP and viral infection.

### 3.4. HSV-1

Three original articles investigated the presence of HSV-1 DNA in OLP tissue biopsies [12,28,57] (Table 3).

Cox et al. [28] examined four case tissues with lichen planus using Southern blot hybridization and found the presence of HSV-1 DNA in two cases (2/4, 50%). Both patients showed HPV16 co-infection. ÓFlatharta et al. [35] decided to employ a nested PCR approach to detect the viral DNA, but they did not retrieve HSV-1 in any of the tested OLP samples (0/26, 0%). Results were also confirmed by Southern blot hybridization. Yildirim et al. [12] decided to detect viral DNA immunohistochemically: they found positivity for the HSV-1 in 6 of the 65 OLP lesions studied (9%). The authors did not find any correlation between HSV positivity and localization, clinical type, gender, or the age of the patients.

### 3.5. CMV

None of the included studies investigate the correlation between CMV infection in OLP tissue specimens.

## 4. Discussion

Numerous studies have tried to find a possible association between OLP and several viruses, trying to identify the actual prevalence of viral infection and to understand if a specific viral infection may play a role in the etiopathogenesis of the disease and the hypothetical malignant transformation of OLP lesions [10,67]. In the present review, HPV-DNA showed a notable prevalence, resulting positive in ≥30% of OLP specimens for 12 out of the 33 included papers. The viral genotype most frequently found was HPV16 (retrieved in 138 out of 1129 analyzed OLP specimens). The variance in HPV DNA prevalence among all the studies presented may be attributed to the differences in sensitivity of the applied molecular methods, sampling methodologies, patient characteristics, and the types of studied specimens (e.g., biopsy tissue, oral mucosal brushing, saliva, and blood samples). Instead, EBV has been detected in 180 out of 440 specimens, achieving a significant total prevalence of 40.9% EBV+ OLP specimens among all the 12 included papers. On the other hand, only 8 out of 95 OLP specimens resulted HSV-1 positive, scoring very low prevalences (8.4%). No influence by gender has been observed, and the studies that investigated these risk factors have demonstrated no significant contribution of smoking and alcohol consumption.

Several authors investigated the possible correlations between the clinical characteristics of OLP lesions and viral infection. Viguier et al. [46] suggest a correlation between severe erosive oral lichen planus and HPV infection and conclude with an interesting theory: they suppose that the HPV16 vaccine may play a protective role against cervical cancer but also towards erosive OLP [68]. Jontell et al. [24], who mainly studied erosive and atrophic OLP lesions, hypothesized that a reduced prevalence of HPV DNA found in these specimens, with respect to other studies on the reticular type of OLP, might be due to a higher degree of keratosis in the reticular type of OLP lesions, a hypothesis supported by the concept that viral replication may require keratinization. Moreover, the most recent studies using the latter techniques could detect HPV DNA more frequently in atrophic-erosive OLP lesions [43]. However, many authors found a similar prevalence of HPV DNA in histologically normal oral lichen planus and oral carcinoma biopsies [23,36], plowing through the line of research on the relationship between malignant transformation in OLP and viral infection. Studies that also included cancer patients in their analysis showed that the majority of oral cancer patients are more likely to be HPV negative but with a history of tobacco and alcohol consumption [13] even if results are equivocal, as other analyses showed a higher prevalence of HPV infection in oral cancer patients [33]. Oddly, Mattila et al. [41] showed no infection of “high risk” HPV16 and 18 genotypes in five OLP lesions that progressed to oral carcinoma. At the same time, they detected “low risk” genotypes 6 and 11 infections in two of these lesions. 

There is still a lack of evidence that HPV could be involved in the etiopathogenesis of OLP, either in the clinical manifestation or the malignant transformation of the lesions.

Some epithelial cancers, instead, have shown a strong association with EBV infection (e.g., nasopharyngeal carcinoma and a subset of gastric and lymphoepithelioma-like cancers) [69]. However, the results obtained in this review are very heterogeneous, with the prevalence of EBV presence in OLP specimens ranging from a minimum of 0% to a maximum of 74%. Even if an abnormal EBV immune status has been demonstrated [54,58], suggesting a possible role of EBV infection and autoimmune dysregulation in OLP patients, current evidence does not support a direct role for EBV in OLP etiopathogenesis, either in malignant transformation or clinical presentation. Prospective cohort studies focusing on a long-term follow-up of OLP patients positive for EBV should be performed to assess its potential malignancy.

Only three studies investigated the prevalence of HSV-1 DNA in OLP lesions, finding very different values ranging from 0% [35] to 50% [28]. However, these studies involved a tiny group of patients with various viral detection techniques that cannot provide significant evidence of HSV-1 involvement in OLP etiopathogenesis or malignant potential.

## 5. Conclusions

In conclusion, the analysis of the 43 selected articles could not confirm a direct involvement in the etiopathogenesis of oral lichen planus or its possible malignant transformation by any of the investigated viruses.

The first point is mainly due to the variance in viral prevalence among all the studies. For the second, however, only two studies [41,47] reported information on how many cases of lichen went on to malignant degeneration and whether viral expression was present in these cases.

However, the results obtained do not allow this correlation to be ruled out either. Indeed, most of the recent studies have shown preliminary results of interest that should be further investigated by conducting long-term follow-up longitudinal studies to obtain results that can be statistically comparable. It would be desirable for future studies to produce comparative statistical analyses and improve the quantity and quality of results to promote the translation of research into clinical practice.

## Figures and Tables

**Figure 1 jcm-11-05487-f001:**
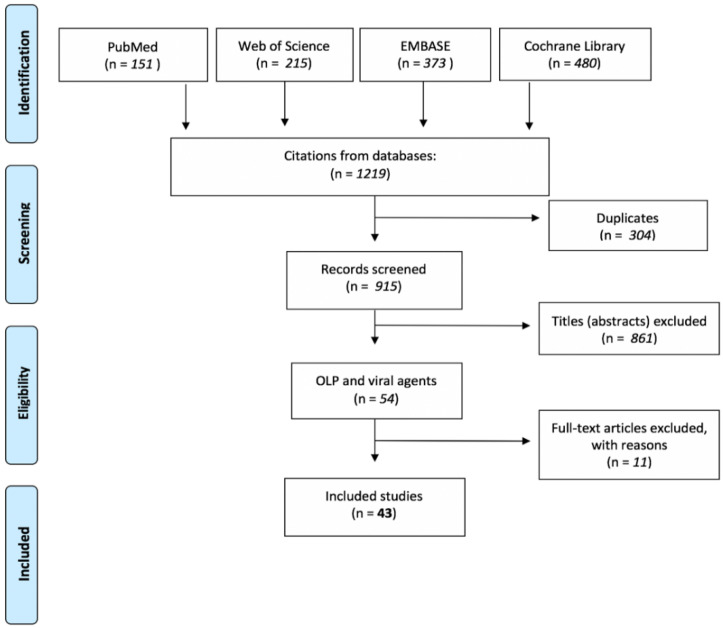
The flowchart explicates the steps made during the selection process.

**Table 1 jcm-11-05487-t001:** The entire list used in the search and the combinations used in the research phase.

Search Topic			
1	OR 2	OR 3	AND 4
			AND 5
			AND 6
			AND 7
			AND 8
			AND 9
			AND 10

1. Oral lichen planus; 2. lichen planus; 3. OLP; 4. human papillomavirus; 5. HSV 1; 6. herpes simplex virus 1; 7. EBV; 8. Epstein–Barr virus; 9. cytomegalovirus; 10. CMV.

**Table 2 jcm-11-05487-t002:** Human papillomavirus (HPV) detection in OLP patients.

Reference (First Author + Year)	Detection of HPV-DNA in Specimens of Oral Lichen Planus (%)	Technique	HPV Probe Used	Specimens Positive for Each Genotype * (n°)	Clinical Features	Malignant Transformation
Syrjänen 1986 [22]	2/2 (100)	ISH	6, 11, 16	11 (1); 16 (1)		

Maitland 1987 [23]	7/8 (87.5)	SBH	1, 2, 4, 6, 11, 13, 16, 18	16 (6); 1 more specimen was positive for HPV but not for the tested specific primers	Reticular keratosic form	
Jontell 1990 [24]	6/20 (30)	SBH	6, 11, 16, 18	11 (6)	Atrophic form with no or mild dysplasia	
Jontell 1990 [24]	13/20 (65)	PCR	6, 11, 16, 18	6 (5); 11 (8); 16 (3)	Atrophic form with no or mild dysplasia	
Kashima 1990 [25]	0/22 (0)	ISH	6, 11, 16, 18, 31	NA	HPV positive OLPs were not classified as erosive form (NS)	
Kashima 1990 [25]	4/22 (18.1)	PAP	6, 11, 16, 18, 31	NA	HPV positive OLPs were not classified as erosive form (NS)	
Young 1991 [26]	0/6 (0)	ISH	6, 11, 16, 18, 31, 33, 35	NA		
Miller 1993 [27]	0/10 (0)	ISH	6, 11, 16, 18, 31, 33, 35, 42, 43, 44, 45, 51, 52, 56	NA		
Cox 1993 [28]	3/4 (75)	SBH	16	16 (3)		
Wen 1997 [29]	2/9 (22.2)	PCR, SBH	16, 18	16 (1); 18 (1)		

González-Moles 1998 [30]	2/17 (11.8)	PCR	16	16 (2)	2 HPV16+ specimens were erosive OLP type	
Sand 2000 [31]	6/22 (27.3)	PCR, SBH	6, 11, 16, 18	18 (5); 1 more specimen was positive for HPV but not for the tested primers	No correlation between the type of OLP and HPV positivity was found	
Yaltirik 2001 [32]	0/2 (0)	ISH	6, 11, 16, 18, 31, 33, 51	NA		
Giovannelli 2002 [33]	9/34 (26.4)	PCR	6, 16, 18, 31, 33	NA		
Ostwald 2003 [34]	10/65 (15.4)	PCR, SBH	6, 11, 16, 18	6 (6); 11 (6); 16 (2); 18 (5)		
ÓFlatharta 2003 [35]	20/38 (52.6)	PCR, SBH	4, 8, 14, 16, 20, 23, 37, 38	4 (1); 8 (2); 14 (1); 16 (11); 20 (1); 23 (1); 37 (1); 38 (7)		
Campisi 2004 [36]	14/71 (19.7)	PCR	6, 16, 18, 31	6 (1); 16 (2); 18 (10); 31 (1)	9 out of 14 HPV positive OLPs were clinically identified as atrophic/erosive variant, and 5 as non-atrophic/erosive variant	
Giovannelli 2006 [37]	13/49 (26.5)	PCR	6, 16, 18, 33, 53	NA	No association was found between type of lesion and site keratinization	
Zarei 2007 [38]	2/15 (13.3)	PCR	6, 11, 16, 18	6 (2)		
Debanth 2009 [39]	NA	NAH	16, 18, 31, 33, 35, 39, 45, 51, 52, 56, 58, 59, 68	NA		
Razavi 2009 [40]	9/29 (31)	PCR	18	18 (9)		
Yildirim 2011 [12]	14/65 (21.5)	ABC	16	16 (14)	2 erosive, 5 plaque, and 7 reticular types of OLP	
Mattila 2012 [41]	13/82 (15.9)	PCR	6, 11, 16, 18, 26, 31, 33, 35, 39, 42, 43, 44, 45, 51, 52, 53, 56, 58, 59, 66, 68, 70, 73, 82	6 (1); 11 (3); 16 (7); 31 (1); 33 (1); 58 (1); 66 (1)		Five patients with atrophic OLP developed invasive OSCC during the follow-up; LR-HPV was detected in two samples: one had HPV6 and the other HPV11
Arirachakaran 2013 [42]	1/37 (2.7)	PCR	6, 11, 16, 18, 30, 31, 32, 33, 34, 35, 39, 40, 42, 44, 51, 52, 53, 55, 56, 58, 59, 66, 68, 70, 71, 73, 74, 81, 82, 85, 90, 91	16 (1)	The only HPV-DNA positive sample came from an atrophic-type OLP lesion	
Kato 2015 [43]	83/200 (41.5)	PCR, ISH, IHC	1, 6, 11, 16, 18, 30, 31, 33, 34, 35, 39, 40, 42, 43, 44, 45, 51, 52, 54, 56, 58, 61, 66, 81	6 (11); 11 (13); 16 (51); 18 (47); 33 (7)	HPV16 positive rates according to OLP clinical type were 28.3% (13/46), 18.6% (8/43), 25.9% (7/27), and 0.0% (0/6) for erosive, reticular, plaque-type, and atrophic	
Pol 2015 [44]	21/30 (70)	IHC	16	16 (21)	All cases of OLP were of the reticular form	
Sahebjamiee 2015 [45]	11/40 (27.5)	PCR	16, 18	16 (5); 18 (3); 3 more specimens were positive for HPV but not for the tested-specific primers		
Viguier 2015 [46]	3/6 (50)	PCR	16	16 (2); 1 more specimen was positive for HPV but not for the tested-specific primers	All cases of OLP were of the severe erosive form	
Liu 2018 [47]	33/46 (71.7)	IHC, OD	16, 18	NA		MT-OLP showed significantly higher infection by HPV16/18(E6) than OLP (*p* < 0.05). HPV infection rates in normal mucosa, OLP, and MT-OLP were 62.50%, 67.50%, and 100.00%, respectively
Zendeli-Bedjeti 2017 [48]	7/31 (22.6)	PCR	16, 18, 31, 56	56 (4); 16 (3)		
Gomez-Armayones 2018 [49]	1/41 (2.4)	IHC, PCR, OD	16, 18	16 (1)		
Sameera 2019 [50]	13/15 (86.6)	PCR	18	18 (13)	Non-atrophic (8/13) and atrophic (5/13) HPV18+ OLP type	
Farhadi 2020 [51]	8/32 (25)	PCR	7, 18	7 (7); 18 (1)	The 8/32 OLP samples that were HPV positive presented erosive features	
Kaewmaneenuan 2021 [52]	11/59 (18.6)	PCR	16, 18	16 (1); 18 (10)	11/59 OLP samples that tested positive for HPV were atrophic/ulcerative type	
Della Vella 2021 [53]	9/52 (17.3)	PCR	6, 8, 11, 16, 18, 26, 31, 33, 35, 39, 40, 42, 43, 44, 45, 51, 52, 53, 54, 56, 58, 59, 61, 66, 68, 69, 70, 73	6 (5); 11(3); 16(1); 42 (2); 53 (1)	8 out of 43 hyperkeratotic OLP type were HPV+; 1 out of 9 erosive OLP type was HPV+ (9 out of 52)	

ISH = in situ hybridization; SBH = Southern blot hybridization; PCR = polymerase chain reaction; NAH = nucleic acid hybridization; IHC = immunohistochemistry; ABC = avidin-biotin-peroxidase complex; PAP = peroxidase-antiperoxidase; OD = optometric density; NA = not available. * In parenthesis is the number of positive specimens for that genotype.

**Table 3 jcm-11-05487-t003:** Detection of other viruses, included in the review, in OLP patients.

Reference (First Author + Year)	Detection of EBV in Specimens of Oral Lichen Planus (%)	Detection of HSV1 in Specimens of Oral Lichen Planus (%)	Detection of HHV7 in Specimens of Oral Lichen Planus (%)	Technique	Clinical Features	Malignant Transformation
Cox 1993 [28]		2/4 (50)		SBH		

Pedersen 1996 [54]	NA (serum analysis)			OD		
Cruz 1997 [55]	1/2 (50)			PCR, SBH		
Sand 2002 [13]	6/23 (26)			nPCR	2 atrophic, 3 reticular, 1 unspecified OLP type (6/23 EBV+)	
ÓFlatharta 2003 [35]	2/26 (7.69)	0/26 (0)	3/26 (11.5)	PCR, SBH		
Sahebjamee 2007 [56]	0/22 (0)			PCR		
Kis 2009 [57]	54/116 (46.5)			nPCR, IHC	26/57 non- atrophic/erosive OLP EBV+ (57/116); 28/59 atrophic/erosive -OLP EBV+ (59/116)	
Yildirim 2011 [12]	23/65 (35.3)	6/65 (9.2)		ABC		
Adtani 2015 [58]	NA (serum analysis)			EI		
Vieira Rda 2016 [59]	15/24 (62.5)			nPCR	9/11 were atrophic/erosive lesions EBV+ (11/24); 6/13 were non-atrophic/erosive lesions EBV+ (13/24)	
Danielsson 2018 [60]	0/25 (0)			ISH, IHC		
Raybaud 2018 [61]	73/99 (74)			ISH, PCR	19/33 were reticular white lesions EBV+ (33/99); 55/66 erosive/ulcerative lesions with or without reticular lesions were EBV+ (66/99)	
Shariati 2018 [62]	6/38 (15.8)			PCR		

SBH = Southern blot hybridization; PCR = polymerase chain reaction (n = nested); NAH = nucleic acid hybridization; IHC = immunohistochemistry; OD = optometric density; ABC = avidin-biotin-peroxidase complex; EI = enzyme immunoassay; NA = not available; ISH = in situ hybridization; WBH = Western blot hybridization.

## Data Availability

Not applicable.

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
