# Peer review of "Correlation between Oral Lichen Planus and Viral Infections Other Than HCV: A Systematic Review"

_jcm, 2022, doi:10.3390/jcm11185487_

Round 1

Reviewer 1 Report

Tha manuscript is well-written and highlights a correct methodology.

Below my few comments:

Delete Table 2, adding a sentence in the text

The "conclusion" section needs to be expanded; in line 294, probably authors intended "long-term follow-up LONGITUDINAL studies instead of  RTCs which are exclusively related to comparison among drugs, medical devices, surgical procedures etc. Please check this finding.

Author Response

Reviewer 1

Tha manuscript is well-written and highlights a correct methodology.

Below my few comments:

Delete Table 2, adding a sentence in the text.

We thank the reviewer for this suggestion. We have removed Table 2 and added the subsection "3.1. Risk of bias in individual studies."  Tables 3 and 4 have become Tables 2 and 3, respectively.

The "conclusion" section needs to be expanded;

Thanks for the suggestion. The Conclusion section has been improved following the reviewer’s comments.

in line 294, probably authors intended "long-term follow-up LONGITUDINAL studies instead of  RTCs which are exclusively related to comparison among drugs, medical devices, surgical procedures etc. Please check this finding.

Thanks for the suggestion. We replaced RTCs with "long-term follow-up LONGITUDINAL studies".

Reviewer 2 Report

The authors present a comprehensive overview of the prevalence and clinical features of the viruses (except for the HCV) in Oral Lichen Planus lesions. They further evaluated whether there is any relationship between malignant transformation in oral lichen planus and viral infection. This is a challenging topic for which there is no comprehensive review. The authors have performed an extensive literature search and summarized the most recent and relevent papers on the subject. Although there are several typos and the narrative is a kind of fragmentary, I just abide by scientific soundness. The text is well written and very easy to read and follow it. I would like to offer the following points for consideration by the authors towards the improvement of the manuscript:

1) Why did not this paper apply the methods of meta-analysis to give a systematic

review and some quantitative analysis on the selected 43 studies?

2) This review paper did not give any clear review results or findings. This paper tends to show that it was not possible to confirm a direct involvement in Oral Lichen Planus etiopathogenesis nor its feasible malignant transformation, Is this the final point of this paper?

3) P7 Line 177 : Please rephrase this sentence “no dysplasia was current”.

4) #4 is not given in the Table 1. Why didn't you use HPV as MeSH term?

5) Please briefly discuss the correlation between the HCV and OPV in Discussion.

Author Response

Reviewer 2

The authors present a comprehensive overview of the prevalence and clinical features of the viruses (except for the HCV) in Oral Lichen Planus lesions. They further evaluated whether there is any relationship between malignant transformation in oral lichen planus and viral infection. This is a challenging topic for which there is no comprehensive review. The authors have performed an extensive literature search and summarized the most recent and relevent papers on the subject. Although there are several typos and the narrative is a kind of fragmentary, I just abide by scientific soundness. The text is well written and very easy to read and follow it. I would like to offer the following points for consideration by the authors towards the improvement of the manuscript:

1) Why did not this paper apply the methods of meta-analysis to give a systematic review and some quantitative analysis on the selected 43 studies?

We thank the reviewer for this question, which allows us to clarify better how we proceeded methodologically in producing this paper.

The 43 selected studies had high heterogeneity in terms of both the methods used (e.g., virus detection techniques) and the results presented (e.g., a large proportion of the studies did not specify the clinical form of lichen).  Because of this heterogeneity, it was preferred to proceed with a systematic review that could provide an overview of all data in the literature to date regarding this topic.

2) This review paper did not give any clear review results or findings. This paper tends to show that it was not possible to confirm a direct involvement in Oral Lichen Planus etiopathogenesis nor its feasible malignant transformation, Is this the final point of this paper?

 We thank the reviewer for this consideration, which allows us to better clarify the concluding part of our paper. Thus, we have improved The Conclusion section, following the reviewer's comment.

3) P7 Line 177 : Please rephrase this sentence “no dysplasia was current”.

Thanks for the suggestion. We rephrased the sentence.

4) #4 is not given in the Table 1. Why didn't you use HPV as MeSH term?

Thanks for the precious suggestion. We added #4 in the table 1.

5) Please briefly discuss the correlation between the HCV and OPV in Discussion.

Regarding this point, we did not feel we would go into more detail regarding HCV as it was not a subject analyzed in the study. Certainly, both HCV and HPV are oncogenic viruses that could play a role in the potential malignant transformation of oral lichen. Given the insufficient data found in the 43 articles analyzed, it might be helpful to explore this correlation in further study.

Round 2

Reviewer 2 Report

I am satisfied that the authors have addressed all of my previous concerns about the article. It is now much improved and I feel that it is now suitable for publication.